# Genetic Modeling of the Neurodegenerative Disease Spinocerebellar Ataxia Type 1 in Zebrafish

**DOI:** 10.3390/ijms22147351

**Published:** 2021-07-08

**Authors:** Mohamed A. Elsaey, Kazuhiko Namikawa, Reinhard W. Köster

**Affiliations:** 1Division of Cellular & Molecular Neurobiology, Zoological Institute, Technische Universität Braunschweig, Spielmannstraße 7, 38106 Braunschweig, Germany; m.elsaey@tu-bs.de; 2Zoology Department, Faculty of Science, Suez Canal University, Ismailia 41522, Egypt

**Keywords:** spinocerebellar ataxia type 1, cerebellum, polyglutamine disease, neurodegeneration, Purkinje cells, bioimaging, zebrafish

## Abstract

Dominant spinocerebellar ataxias (SCAs) are progredient neurodegenerative diseases commonly affecting the survival of Purkinje cells (PCs) in the human cerebellum. Spinocerebellar ataxia type 1 (SCA1) is caused by the mutated *ataxin1* (*Atx1*) gene product, in which a polyglutamine stretch encoded by CAG repeats is extended in affected SCA1 patients. As a monogenetic disease with the Atx1-polyQ protein exerting a gain of function, SCA1 can be genetically modelled in animals by cell type-specific overexpression. We have established a transgenic PC-specific SCA1 model in zebrafish coexpressing the fluorescent reporter protein mScarlet together with either human wild type Atx1[30Q] as control or SCA1 patient-derived Atx1[82Q]. SCA1 zebrafish display an age-dependent PC degeneration starting at larval stages around six weeks postfertilization, which continuously progresses during further juvenile and young adult stages. Interestingly, PC degeneration is observed more severely in rostral than in caudal regions of the PC population. Although such a neuropathology resulted in no gross locomotor control deficits, SCA1-fish with advanced PC loss display a reduced exploratory behaviour. In vivo imaging in this SCA1 model may help to better understand such patterned PC death known from PC neurodegeneration diseases, to elucidate disease mechanisms and to provide access to neuroprotective compound characterization in vivo.

## 1. Introduction

Spinocerebellar ataxia type 1 (SCA1) is an adult onset, autosomal dominant neurodegenerative disease, in which Purkinje cells (PCs) of the cerebellum are among the most prominent neuronal populations that are affected by progressive cytotoxicity and cell death [1]. Being the sole output neurons of the cerebellum, progressive loss of PCs results in progredient impairment of cerebellar functions. Therefore, in SCA1 patients motor coordination, visuospatial performance, psychiatric health and cognitive abilities successively decline leading to death within about 15 years after the first appearance of symptoms [2,3]. In the past, a number of highly informative and elegant studies have been performed to reveal the underlying mechanisms resulting in the SCA1 pathology [4,5], but the knowledge of the cellular and molecular mechanisms resulting in SCA1-mediated neurodegeneration is still incomplete. Therefore, no cure exists for this disease, and treatment is limited to symptom amelioration.

SCA1 is an inherited monogenetic disease largely caused by gain of function of the human Ataxin1 protein (Atx1) [6]. This protein contains a polyglutamine (polyQ) stretch of variable length in its N-terminus, with non-interrupted polyQ-stretches of above 40 glutamine residues leading to SCA1 symptoms and neurodegeneration [7,8]. The length of this uninterrupted stretch inversely correlates with the age of onset, severity of pathogenesis, and progression of SCA1 [9]. At least for Huntington’s disease, another neurodegenerative polyglutamine disease, it was shown that interruption of the polyQ-stretch significantly decreases toxicity, and that the length of the non-interrupted polyQ-stretch rather than its overall length determines the onset of the disease [10]. Similarly, SCA1-patients typically contain non-interrupted polyQ-stretches in their *Atx1* allele, while the polyQ-stretch in non-pathological alleles is often found to be interrupted by histidine residues. Such histidine insertions into the polyQ-stretch have been proposed to decrease the aggregation rate of the Atx1 protein and to result in different types of aggregates of amyloidic nature [11]. These aggregates are less stable and are therefore easier to remove by the cell preventing proteotoxic cell death. While the wild type Atx1 protein is present in both the cytoplasm and the nucleus, the pathological polyglutamine containing mutant Atx1-polyQ accumulates in the nucleus, where it can aggregate into nuclear inclusion bodies. The nuclear accumulation of mutant Atx1-polyQ is a prerequisite for disease onset and neurotoxicity [12,13].

The current knowledge about cellular and molecular mechanisms underlying SCA1 are largely derived from modelling this disease in fruit fly and mouse and exploiting the specific advantages of these animal models [14,15]. For example, the retina degeneration model in *Drosophila* has provided rapid screening for genes involved in mediating Atx1-polyQ-induced cell death [16,17]. Transgenic mouse models have been a powerful tool to address molecular mechanisms of PC degeneration and the resulting behavioural defects in the SCA1-affected mice, with the observed phenotypes being similar to those of human patients [12,15,18].

While in *Drosophila* SCA1 needs to be modelled in heterotypic cell populations due to the lack of a cerebellum in invertebrates, mice find their limitations in longitudinal in vivo studies such as in vivo time-lapse microscopy of dynamic cell biological or physiological processes. Therefore, genetic models of SCA1 in PCs of a vertebrate accessible for high resolution in vivo microscopy, behavioural and pharmacological investigations would be valuable and likely informative.

Zebrafish are genetically tractable; their larvae are small and nearly transparent with the unfolded zebrafish cerebellum being positioned directly underneath the skin of the dorsal hindbrain. These advantages make zebrafish nearly ideal for cell biological and physiological studies directly in vivo [19,20,21,22]. In addition, the cerebellum is highly conserved among vertebrates such that in teleosts nearly all cell types of the human cerebellum can be found displaying a highly conserved morphology, connectivity, and function [23].

The zebrafish genome contains two *ataxin1* homologs encoding for proteins with conserved functional domains essential for SCA1 toxicity, and immunohistochemical studies have shown that zebrafish *ataxin1* gene products are expressed in cerebellar PCs [24]. In addition, we have recently developed an approach to genetically target differentiating and mature zebrafish PCs with co-expression of several transgenes [25]. We have used this tool to establish a genetic model of human SCA1 in zebrafish for observing affected PCs using fluorescent protein reporters and to monitor the behavioural performance of affected fish in relation to their disease progression.

## 2. Results

### 2.1. Genetic Modeling of SCA1 in Zebrafish

We have recently demonstrated that a small genomic element upstream of the zebrafish gene *carbonic anhydrase 8* (*ca8*), named cpce, is able to exclusively drive expression in zebrafish PCs starting at their timepoint of differentiation, which initiates roughly at 2.5 days postfertilization (dpf) [25]. Adding two times four copies of this regulatory element in opposite orientation and flanking it with basal promoters on both sides mediates robust PC-specific expression of two different transgene cassettes (Figure 1A). One of these was used to express a membrane targeted variant of a red fluorescent protein mScarlet, GAPmScarlet. The other was either left empty or expressed a cDNA of human *ataxin1* containing a non-pathological polyglutamine stretch of 30 glutamine residues Atx1[30Q] interrupted by two histidine residues (both of these constructs served as controls), while a patient-derived pathological mutant with 82 non-interrupted glutamine residues, Atx1[82Q], served to model SCA1 (Figure 1A). Both of these alleles were fused to an HA-tag at their N-terminus for immunohistochemical detection of the respective Atx1 proteins.

These constructs were microinjected into one-cell stage zebrafish embryos together with mRNA encoding the transposase Tol1 to promote genomic integration and germline transmission of the respective transgene cassettes early during development [26]. At 4 dpf the injected larvae were screened and those found to have broad GAPmScarlet expression throughout the PC population were raised to adulthood (P0 generation). Germline transmission was examined by breeding individual P0 to wild type fish and screening of the filial F1 generation for PC-specific mScarlet fluorescence. Such carriers were raised to adulthood and bred with wild type fish to confirm transmission of the fluorescent reporter to the F2 offspring in a Mendelian ratio (Figure 1B). Based on individual F1 founders, three independent transgenic lines were established.

Transgenic embryos of these SCA1 zebrafish lines, when analysed at 4 dpf by laser scanning confocal microscopy, were found to display continuous non-mosaic expression of the fluorescent reporter GAPmScarlet throughout the PC layer (Figure 1Ca–c). Crossing transgenic fish to carriers of the *Tg(−7.5ca8:GFP)^bz12^* strain with PC-specific expression of EGFP [25] confirmed that transgene expression was confined to the PC population (Figure 1Cd–f). Finally, to prove that GAPmScarlet fluorescence in PCs was accompanied by *Atx1*-allele expression HA-immunohistochemistry was performed. This analysis revealed that hAtx1[30Q] and hAtx1[82Q] were coexpressed in PCs throughout the entire population indicating that GAPmScarlet serves as a reliable intravital marker for PCs expressing human *Atx1* alleles (Figure 2A,B).

### 2.2. Age-Related Progressive Disturbances of Purkinje Cell Layer Integrity in Zebrafish Genetic Model of SCA1

In order to reveal the effect of transgene expression on developing and mature PCs, heterozygous F2 offspring were raised and their red fluorescent cerebellum was analysed by laser scanning confocal microscopy.

Early larval development appeared to occur indistinguishably between control fish expressing red fluorescent membrane targeted GAPmScarlet in PCs, controls coexpressing the human non-pathological allele Atx1[30Q], and zebrafish coexpressing the pathological allele Atx1[82Q]. Specimens carrying any one of the three alleles displayed a continuous bright red fluorescence throughout the PC population. Therefore, transgenic carriers were raised to late larval (40 dpf), juvenile (about 60 dpf, 2 months), and young adult (3 months) ages. At these timepoints, their body length was measured (Figure 3A) followed by subsequent dissection of the brain for microscopy analysis of PCs at further detail (Figure 3B). PCs still expressed the red fluorescent reporter protein confirming the continued expression of the transgene driven by cpce throughout adulthood as demonstrated previously [25].

At 40 dpf PCs in GAPmScarlet control (Figure 3Ca), Atx1[30Q] (Figure 3Cb), and Atx1[82Q] (Figure 3Cc) heterozygous transgene carriers formed a continuous layer covering the corpus cerebelli with both cerebellar hemispheres being separated by the dorsal midline (Figure 3C white dashed line). No major signs of cerebellar atrophy, like shrinkage in tissue or discontinuous patches of fluorescent PCs, could be identified at this developmental age. Furthermore, the body length between individuals from all three allelic families did not differ significantly, and this continued growth in body length remained until adulthood (Figure 3D). This suggests that despite the expression of the pathological Atx1[82Q] SCA1-causing allele in cerebellar PCs, zebrafish do not seem to suffer from major locomotive impairments interfering with proper feeding. The absence of a major cerebellar atrophy in all 40 dpf samples was further confirmed when the area covered by fluorescent PCs was quantified from maximum projections derived from z-stacks of confocal microscopy images and compared among specimens from all three allelic families, which revealed no significant size differences (Figure 3G). Nevertheless, in contrast to controls and Atx1[30Q] zebrafish, confocal microscopy of the PC population in Atx1[82Q] expressing larvae depicted speckles of brighter fluorescence suggestive of condensation, cellular shrinkage and onset of neurodegeneration (Figure 3Cd–f, compare PCs marked by white arrows).

In juvenile fish of two months of age GAPmScarlet (Figure 3Ea) and Atx1[30Q] (Figure 3Eb) control fish displayed a continuous PC population marked by red fluorescence in both hemispheres, which had spread in size (Figure 3G) together with an increase in body length (Figure 3D). This increase in body length accompanied by an age-related expansion of the PC population in the corpus cerebelli continued to young adulthood in three month old fish (Figure 3D,G), with a PC population displaying a dense continuous layer of red fluorescent cells (Figure 3Fa,b).

In contrast to this, 2 month old juvenile Atx1[82Q] fish showed irregularities in their fluorescent PC population with signs of neurodegeneration based on the observance of areas of decreased or even absent red fluorescence (Figure 3Ec–f compare PCs marked by white arrows). This impaired age-dependent expansion of the PC population became obvious when the size of the PC area was determined, which had barely increased in comparison to 40 dpf old larvae (Figure 3G) despite a regular age-related increase in body size (Figure 3D). In young adults at three months of age, this situation progressed even further. Although the fish had grown to their normal body size compared to wild type and Atx1[30Q] counterparts (Figure 3D), the PC population in Atx1[82Q] carriers was significantly disrupted with a severely disturbed outline, large discontinuities of cells, a small size, and a granular appearance (Figure 3Fc). The area covered by the PC population was quantified to be smaller than in 40 dpf larvae and about five-fold smaller (0.11 mm^2^) than in the control counterparts (GAPmScarlet and Atx1[30Q] 0.47 mm^2^ and 0.57 mm^2^, respectively) (Figure 3G). Confocal microscopy of selected PC areas at higher magnification (Figure 3Fa–c, white squares) revealed condensed PC somata of unusually bright fluorescence likely caused by cell shrinkage and condensation of the fluorophore (Figure 3Ff, white arrows) and fluorescent cellular debris indicative of an ongoing degeneration of PCs, while in GAPmScarlet (Figure 3Fd) and Atx1[30Q] (Figure 3Fe) carriers a fine web of neuropil formed by PC dendrites was visible.

These findings suggest a progredient degeneration of PCs in the zebrafish PC-specific SCA1 model expressing the human Atx1[82Q] allele, which like in human patients occurs in a progressive, age-related manner. While signs of neuronal degeneration can be seen already in the larval stages, changes in the size of the PC population become evident in juveniles as has been reported for human patients carrying pathological Atx1 alleles with anticipation of the disease at juvenile ages [27].

### 2.3. Progredient PC Degeneration in Zebrafish Genetic Model of SCA1

To further corroborate that PCs in the Atx1[82Q] SCA1 zebrafish model specifically suffer from a progredient degeneration, sagittal sections from the cerebellum of controls, Atx1[30Q] and Atx1[82Q] heterozygous adult fish at three months of age were stained with hematoxilin and eosin. This histological staining confirmed the largely undisturbed cytology of the cerebellum with a clearly distinguishable granule cell layer (GCL) and molecular layer (ML) consisting mostly out of neuropil (Figure 4a–c).

At higher magnifications the Purkinje cell layer (PCL) containing cells with characteristic large somata could be observed in control and Atx1[30Q] cerebelli (Figure 4d,e white arrows), yet such somata were rarely found in the section of the cerebellum from Atx1[82Q] carriers (Figure 4f). Indeed, membrane targeted expression of mScarlet revealed a continuous cell layer of PCs in sagittal sections in GAPmScarlet and Atx1[30Q] controls (Figure 4g,h), while in Atx1[82Q] SCA1 fish red fluorescent PCs were largely absent, intriguingly almost entirely in anterior regions of the corpus cerebelli, while the posterior region still contained a number of PCs (Figure 4i). This observation was further confirmed by immunohistochemistry against the PC-specific ZebrinII, which in contrast to GAPmScarlet and Atx1[30Q] carriers (Figure 4k,l) revealed a disrupted organization of the PC layer in the corpus cerebelli of Atx1[82Q] SCA1 fish with an apparent absence of PCs that was more pronounced in anterior than in posterior regions (Figure 4l). Finally, higher magnifications of sagittal sections of the PC layer in Atx1[30Q] (Figure 4m) and Atx1[82Q] (Figure 4n) heterozygous fish analysed by anti HA-tag immunohistochemistry confirmed that the transgenes were still expressed in PCs during adulthood, yet displaying only sparse labelling in PCs of Atx1[82Q] carriers consistent with apparent holes in the PC layer, cellular debris and pronounced PC degeneration. These findings further support the age-dependent progredient loss of PCs in the Atx1[82Q] SCA1 fish as it is observed in SCA1 patients. Furthermore, these results confirm an intriguing observation from juvenile stages that apparently anterior regions of the corpus cerebelli seem more vulnerable to PC degeneration than posterior regions in the context of this zebrafish SCA1 model.

### 2.4. Behavioural Performance of Zebrafish SCA1 Model

When zebrafish of our SCA1 model were examined for their swimming properties, we could not detect any obvious differences between control, Atx1[30Q], and Atx1[82Q] specimens. Yet, in the fish facility Atx1[82Q] carriers appeared to stay closer to the bottom of their tanks and seemed less explorative.

To address this observation systematically, we performed a novel tank test assay in which zebrafish are placed into the new environment of a novel tank and are observed during the first six minutes for their exploratory behaviour. This assay addresses the natural tendency of zebrafish to initially dive to the bottom of a novel experimental tank, with a gradual increase in the vertical exploratory activity over time, while longer bottom times are interpreted as increased anxiety behaviour in zebrafish [28]. Individual heterozygous carriers of all three genotypes (GAPmScarlet, Atx1[30Q] and Atx1[82Q]) were tested at 40 dpf, two months, and three months of age to relate the behavioural performance to the observed degree of PC degeneration.

In general, at a younger age of 40 dpf, zebrafish larvae spend more time at the bottom of a novel tank, compared to at two and three months of age, by which time they become more explorative (Figure 5A). Strikingly, while control and Atx1[30Q] specimens indeed showed this age-related increase in exploratory behaviour (Figure 5A,C grey and blue dots, respectively), Atx1[82Q] did not progress in their exploration and remained at the bottom of the novel tank for almost the entire observation period also at 2 and 3 months of age (Figure 5B,C red dots). Although the number of entries into the upper half of the novel tank did not vary with age among all genotypes tested (Figure 5D), the latency to enter the upper half for the first time decreased with age (Figure 5E). This demonstrates that Atx1[82Q] fish did aim to enter the upper half of the tank and became more explorative and thus were able to swim to the upper half, but they barely remained at upper levels and entered the upper half of the tank for only short moments. To further exclude that upper tank half stays in Atx1[82Q] fish are reduced because of locomotive problems, we compared the distance swam over time (Figure 5F), which increased with age similarly among all genotypes. Similarly, the minimum velocity (Figure 5G) and maximum velocity (Figure 5H) of swim movements was not significantly different (with the exception of the minimum velocity of Atx1[30Q] fish at three months, for which we currently have no explanation).

Thus, Atx1[82Q] fish of the established SCA1 zebrafish model do not seem to display severe locomotive impairments in spite of the progressive PC degeneration, but exploratory behaviour typical for their respective age is impaired and exposure to a novel environment is avoided. This observation fits well with the increase in psycho-social problems observed in SCA1 patients [3,29].

## 3. Discussion

With an increase in average age in the worldwide population, the prevalence of age-associated neurodegenerative diseases increases, which poses a severe health, economic, and social burden. Although most cases of progressive neurodegenerative diseases occur sporadically, familial forms with clear patterns of inheritance and identified causative genes provide the opportunity to genetically model neurodegenerative diseases in molecularly tractable animals for elucidating cellular and molecular principles underlying neuronal death. Spinocerebellar ataxias (SCAs) caused by polyglutamine proteins are autosomal dominant diseases of monogenetic origin with the affected protein predominantly exerting a proteotoxic gain of function. This allows for modelling of these diseases by overexpression approaches, and therefore mouse models for SCA1, SCA2, SCA3, SCA6, SCA7, and SCA17 have been established [4]. Studies in these mammalian models have provided a wealth of information about disease mechanisms and the development of possible therapeutic strategies. These models, however, are less well-suited for non-invasive imaging approaches to address, for example, the cell biological mechanisms underlying neurodegeneration directly in vivo. Here, cellular and subcellular dynamics need to be deduced from observations at fixed time points or require studies in cultured tissues and cells.

How could zebrafish as a non-mammalian aquatic vertebrate model contribute to these studies? The external development of zebrafish larvae, small size and optical translucency as well as their fecundity make them well-suited for longitudinal non-invasive microscopy studies, high resolution in vivo cell biology, optogenetic approaches, behavioural analysis, and compound characterization. In addition, zebrafish are genetically tractable and loss of function studies by means of antisense RNA injections called morpholinos have been used in a number of investigations to model SCA diseases in zebrafish. A recent review summarizes the efforts to model autosomal recessive ataxias in zebrafish in detail [30]. As morpholino approaches can suffer from off-target effects, some of the described phenotypes await confirmation by targeted mutagenesis, for which the advent of the CRISPR/Cas9-technology is well suited. In contrast, autosomal dominant ataxias can be modelled in zebrafish by overexpression approaches and the generation of stable transgenic fish, for which transposon-mediated chromosomal integration is a straight-forward technology [26]. Yet, such studies require precise regulatory elements for cell-type specific expression. With respect to spinocerebellar ataxias, only two Purkinje cell specific regulatory elements have recently become available [25,31]. This may explain why gain of function approaches to model autosomal dominant ataxias in zebrafish are currently limited. For most of the pathological genes, the endogenous function has been addressed again by morpholino-mediated approaches, which are summarized well in a recently published review article [32]. With respect to polyglutamine SCA disease models, SCA3 is the only disease for which a zebrafish model currently exists [33].

We have therefore set out to genetically model SCA1 in zebrafish cerebellar Purkinje neurons by expressing a human patient derived variant of the disease-causing gene *ataxin1* containing a non-interrupted stretch of 82 glutamine residues in its N-terminus. Using the human Atx1[82Q] should facilitate the relation of pharmaceutical studies to results in mammalian systems. Restricting the expression of this nuclear protein to a single neuronal cell type will help to unravel its cell-autonomous consequences and allows for relating behavioural changes that emerge from the compromised function of a distinct neuronal population to its degree of degeneration. Here, the coexpression of a fluorescent protein serves as an easy readout for the progression and severity of PC degeneration. Importantly, SCA1 is an adult-onset neurodegenerative disease with progredient PC degeneration and this is reflected in the established zebrafish model in which first signs of PC degeneration are observed during young juvenile stages when PCs have already matured [25,34]. Subsequently, PC degeneration progresses such that within about two months almost all PCs are lost in the zebrafish cerebellum. As PCs are the only output neurons of the cerebellum, significant PC degeneration should result in behavioural deficits. Indeed, we have observed a decrease in exploratory behaviour with the severity correlating to the degree of PC degeneration. Interestingly, malfunction of cerebellar PCs does not seem to lead to a severe disruption in swimming abilities in zebrafish. Neither significant differences in travelled distance nor altered swim velocity could be observed in Atx1[82Q] expressing SCA1 fish compared to Atx1[30Q] or GAPmScarlet expressing controls. It is often observed in neurodegenerative diseases that a substantial amount of neurons can be progressively lost without causing significant symptoms. Therefore, these diseases are commonly diagnosed at a late stage. In addition, remaining PCs still present at three months of age could exert plasticity and take over functions of already degenerated PCs. Furthermore, this observation is in accordance with other zebrafish models in which PCs are impaired. Rather than gross defects in locomotor control, socio-emotional functions such as conditioned active avoidance are observed [35].

Interestingly, in the established zebrafish SCA1 model, PC degeneration did not appear to progress equally in all regions of the corpus cerebelli. In its anterior regions, PC degeneration was predominant, while posterior regions were less severely affected. Recently, several experimental studies have revealed that the PC population in zebrafish is functionally regionalized. While rostro-medial PC populations are involved in processing novel associations and adjustments of swim patterns like turning behaviour, caudal regions of the PC population are assigned to balance control and eye and body coordination reminiscent of the flocculonodulus [20,36]. This might contribute to the observation that exploratory behaviour rather than locomotor control is affected in SCA1 zebrafish. Additionally, in human SCA1 patients PC loss does not occur uniformly, but affects most severely the hemispheres of the cerebellum, the vermis is less affected, while the flocculonodular lobes display the lowest degree of PC degeneration [37,38].

Therefore, the established SCA1 zebrafish provide an intriguing model to study region specific PC contributions to cerebellar neurodegeneration phenotypes. Furthermore, with the possibility for in vivo imaging approaches, these SCA1 zebrafish provide a powerful tool to interconnect molecular and cellular mechanisms of PC degeneration with physiological analysis and behavioural studies to provide longitudinal structure-function-behaviour relationship insights.

## 4. Materials and Methods

### 4.1. Animal Husbandry and Maintenance

Zebrafish were maintained and raised at 28 °C on a 14:10 h light:dark cycle according to standard protocols [39,40]. Natural mating was used to obtain embryos and larvae and staging was performed according to days postfertilization (dfp) [41]. Zebrafish embryos were incubated in egg water (0.03% g/L sea salt) for 6 h and then kept in Danieau medium (0.12 mM MgSO_4_, 0.21 mM KCl, 0.18 mM Ca(NO_3_)_2_, 17.4 mM NaCl, 1.5 mM HEPES, pH 7.2). For immunohistochemistry and live imaging experiments the Danieau medium was supplemented with 0.005% phenylthiourea (PTU, Sigma Aldrich, St. Louis, MO, USA) to suppress pigmentation. The following transgenic zebrafish strains used in this study have been described previously: *Tg(−7.5ca8:GFP)^bz12^* [25].

### 4.2. Plasmid Construction

To generate stable transgenic zebrafish, three expression constructs in the pBuescriptII backbone flanked by Tol1 transposase recognition sites and containing 8xcpce PC-specific regulatory elements flanked by E1b basal promoters, multiple cloning site, 3′UTRs composed of four copies of mir181aT target sites (4xmir181aT) and SV40 polyA sequences were used as described previously [25]. According to the nomenclature of this bidirectional construct, a cDNA encoding for membrane targeted GAPmScarlet red fluorescent protein as *EcoR*V/*Xba*I-fragment was inserted into the left vector arm opened by *EcoR*V/*Spe*I. Membrane targeting of mScarlet is achieved by a palmitoylation signal peptide sequence (MLCCIRRTKPVEKNEEADQE-) derived from zebrafish GAP43 that was fused to the N-terminus of mScarlet. cDNAs encoding either human *Atx1*[30Q], with an interrupted polyQ-stretch [QQQQQQQQQQQQQQHQHQQQQQQQQQQQQQQQ], or SCA1 patient-derived *Atx1*[82Q], with an uninterrupted polyQ-stretch, both N-terminally fused with an HA-tag for detection of protein expression were inserted as *Eco*RI/*Xba*I-fragments into the right vector arm opened with the same restriction enzymes. From this cloning strategy three expression constructs pTol1-8xcpce:GAPmScarlet (#5158), pTol1-GAPmScarlet-8xcpce-hAtx1[30Q] (#5648), and pTol1-GAPmScarlet-8xcpce-hAtx1[82Q] (#5187) were generated (Figure 1A).

### 4.3. Generation of Transgenic Zebrafish

To establish transgenic zebrafish strains, embryos were coinjected with plasmid DNA (25 ng/µL) and mRNA encoding Tol1 transposase (25 ng/µL) at the one-cell stage [26]. Transient transgenic larvae were analysed for fluorescent PCs between 3 and 7 dpf, specimens with broad GAPmScarlet fluorescent protein expression throughout the PC layer were raised to adulthood and screened for stable germline integration in their offspring. For each transgene cassette P0 founders giving rise to offspring with faithful and prominent red fluorescence covering the cerebellar surface were used to establish three independent F1-families. As these showed comparable expression, one transgenic line for each construct was eventually continued, which are currently in the F3-generation.

### 4.4. Histology and Immunohistochemistry

To isolate brains from juvenile and adult zebrafish, 0.04% tricaine was used to anesthetize the fish. It was then cooled down on ice and the length of the body from the anterior tip of the jaw until the caudal indentation of the tail fin (fork length) was measured. Subsequently, the fish was decapitated and the head was transferred to a Petri dish with cold 1× PBS for further manual dissection of the brain under a light microscope. The isolated brain was transferred into cold 4% PFA/PBS and incubated at 4 °C overnight. After washing in cold 1× PBS three times for 5 min. each, the brain was used either for whole brain imaging by confocal laser scanning microscopy (LSM 880 with Airyscan, Carl Zeiss AG, Jena, Germany) or used for sagittal sectioning into 40 µm sections using a vibratome (Leica Biosystems Inc., Wetzlar, Germany). These sagittal sections were stained with Hematoxylin-Eosin, anti-ZebrinII antibody staining, or anti-HA antibody staining. Subsequently, stained sections were visualized by stereomicroscopy.

Whole-mount immunohistochemistry on 4 dpf zebrafish larvae was performed according to a previously published protocol [42]. Anti-HA rat antibody (1:1000, Roche Inc., Basel, Switzerland) was used as the primary antibody for immunostaining of human Atx1 protein. Alexa Fluor-488 goat anti-rat IgG antibody (1:1000, Thermo Fischer Scientific, Carlsbad, CA, USA) was used as a secondary antibody. To detect Zebrin-II by immunohistochemistry a monoclonal mouse anti-ZebrinII antibody was used (1:500, gift from R. Hawkes, Univ. Calgary, AB, Canada), and an Alexa-Fluor-488 goat anti-mouse IgG antibody (1:1000, Thermo Fischer Scientific, Carlsbad, CA, USA) was used as a secondary antibody. The recording of fluorescent images was performed with the confocal laser scanning microscope (LSM 880 with Airyscan, Carl Zeiss AG, Jena, Germany). The images were constructed from z-stack slices with the help of the ZEN software (Carl Zeiss AG, Jena, Germany).

### 4.5. Behavioural Analysis

The novel tank diving test is a behavioural test used to characterize the exploratory behaviour in zebrafish [28]. The novel tank was positioned in front of a video camera for optimal recording. Light from the back provided sufficient contrast for accurately tracking individual fish. Upon start of video recording, individual fish were gently transferred to the novel tank environment [43]. The behaviour of the fish was recorded for six minutes and then it was returned to its home tank. Automated tracking and analysis of swim movements was performed by Ethovision XT 12 software (Noldus Inc., Wageningen, The Netherlands). Data was transferred to Excel sheets (Microsoft Inc., Redmond, WA, USA) for further statistical analysis.

### 4.6. Statistical Analysis

A two-way ANOVA with post hoc Tukey’s test was used to evaluate statistical significance of differences between experimental groups. Data are presented as mean ± SEM; * *p* < 0.05, ** *p* < 0.01, *** *p* < 0.001, and **** *p* < 0.0001. Experimental numbers of the fish per group are reported in the respective figure legends.

## Figures and Tables

**Figure 1 ijms-22-07351-f001:**
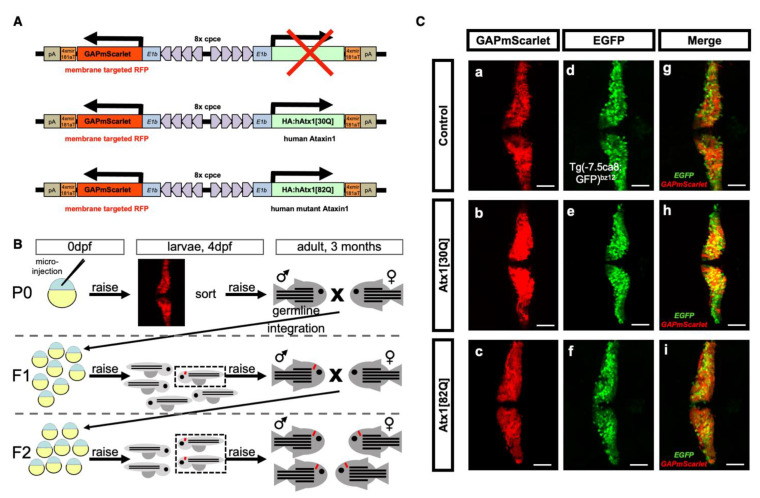
Generation of a stable transgenic zebrafish model for PC-specific SCA1. (**A**) Schematic drawing (not to scale) of expression constructs for PC-specific expression of red fluorescent reporter protein mScarlet targeted to the cytoplasmic membrane due to an N-terminal fusion with a post translational palmitoylation signal derived from the zebrafish GAP43 protein. This reporter is expressed alone, or coexpressed with non-pathological [30Q] and mutant pathological [82Q] alleles of human *ataxin1*. Ectopic expression in tectal neurons and retinal cells was eliminated by four copies of miRNA181a target sites (4xmiR181aT) in the 3′UTR of the transgenes [25]. All bidirectional transgene cassettes are flanked by Tol1 transposase recognition sites [26]. (**B**) Schematic outline of screening and breeding procedure to obtain stable transgenic zebrafish of the F2-generation, which propagate PC-specific mScarlet and Atx1 expression in a Mendelian manner. (**C**) Analysis of GAPmScarlet expression (**a**–**c**) of carriers crossed into the transgenic *Tg(−7.5ca8:GFP)^bz12^* background with PC-specific cytoplasmic EGFP expression (**d**–**f**) and merged images (**g**–**i**) to confirm PC-specific coexpression of the transgene cassette in F2 larvae at 4 dpf of the established transgenic control mScarlet (upper row), Atx1[30Q] (middle row), and Atx1[82Q] (lower row) strains, respectively. The individual images display dorsal views of the larval corpus cerebelli, anterior is to the left. Scale bar: 50 µm.

**Figure 2 ijms-22-07351-f002:**
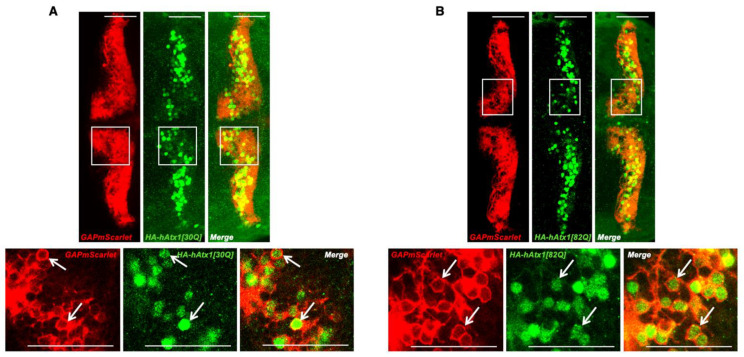
Analysis of human *Atx1* allele expression in the stable transgenic zebrafish strains. Anti-HA immunostaining of HA-tagged proteins hAtx1[30Q] (**A**) and hAtx1[82Q] (**B**) in transgenic 4 dpf F2 larvae of established Atx1[30Q] and Atx1[82Q] strains crossed into the transgenic *Tg(−7.5ca8:GFP)^bz12^* backgroundCytoplasmic and nuclear anti-HA-derived green fluorescence is surrounded by PC-specific membrane targeted red fluorescent GAPmScarlet expression (white arrows) The images display a dorsal view of the larval zebrafish corpus cerebelli, anterior is to the left. While in the upper row an overview over both cerebellar hemispheres is shown, the white boxes depict an area that was displayed at higher magnification in the lower row. Scale bars: 50 µm.

**Figure 3 ijms-22-07351-f003:**
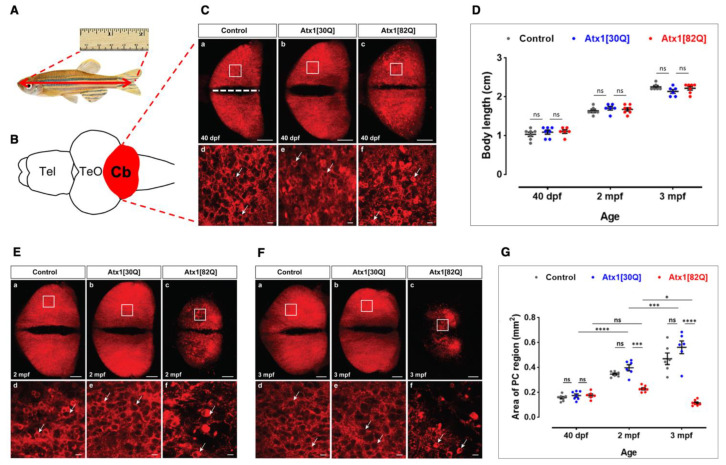
Progredient disintegration of PC layer in genetic model of SCA1 in zebrafish. (**A**) Schematic drawing of adult zebrafish to demonstrate measurement of body length form the anterior tip of the jaw until the caudal indentation of the tail fin. (**B**) The cerebellum in dissected brains from these specimens is easy to identify by the red fluorescent PC population forming a continuous layer of cells across the corpus cerebelli (dorsal view). (**C**) Maximum brightness projection of images stacks recorded by confocal microscopy from control (**a**,**d**), Atx1[30Q] (**b**,**e**), and Atx1[82Q] (**c**,**f**) expressing heterozygous carriers of the respective transgene at 40 dpf. Note speckles of brighter fluorescence suggestive of condensation and cellular shrinkage can be observed in the PC layer of Atx1[82Q] carriers not found in GAPmScarlet and Atx1[30Q] controls (compare white arrows). (**D**) Body length measurements reveal a similar overall growth rate to control (*n* = 6), Atx1[30Q] (*n* = 6), and Atx1[82Q] (*n* = 6) fish. (**E**) Maximum brightness projection of image stacks recorded by confocal microscopy from control (**a**,**d**), Atx1[30Q] (**b**,**e**), and Atx1[82Q] (**c**,**f**) expressing heterozygous carriers of the respective transgene at two months of age and (**F**) three months of age, respectively. The area marked by a white rectangle is displayed at higher magnification below for each specimen, respectively. Compared to controls and Atx1[30Q], young adult Atx1[82Q] zebrafish at three months of age display a clear atrophy of their PC layer containing widespread fluorescent debris of degenerated PCs (Ff white arrows). (**G**) Quantification of area covered by fluorescent PCs, while in controls (*n* = 6) and Atx1[30Q] carriers (*n* = 6) a continuous age-dependent increase of the PC layer occurs, the expansion of the PC layer in Atx1[82Q] carriers (*n* = 6) stalls. (**C**,**E**,**F**): dorsal views of corpus cerebelli, anterior to the left, scale bars: 100 µm (**a**–**c**) or 10 µm (**d**–**f**). The data in (**D**,**G**) are presented as mean ± SEM, * *p* < 0.05, *** *p* < 0.001 and **** *p* < 0.0001 according to two-way ANOVA with post hoc Tukey’s test. *n* = 7, 6, 6 in 40 dpf, 2 mpf (months postfertilization) and 3 mpf control groups, respectively. *n* = 7, 6, 6 in 40 dpf, 2 mpf, and 3 mpf Atx1[30Q] groups; and *n* = 6, 7, 7 in 40 dpf, 2 mpf, and 3 mpf Atx1[82Q] groups, respectively. Abbr.: Cb: cerebellum, PC: Purkinje cell, TeO: optic tectum, Tel: telencephalon.

**Figure 4 ijms-22-07351-f004:**
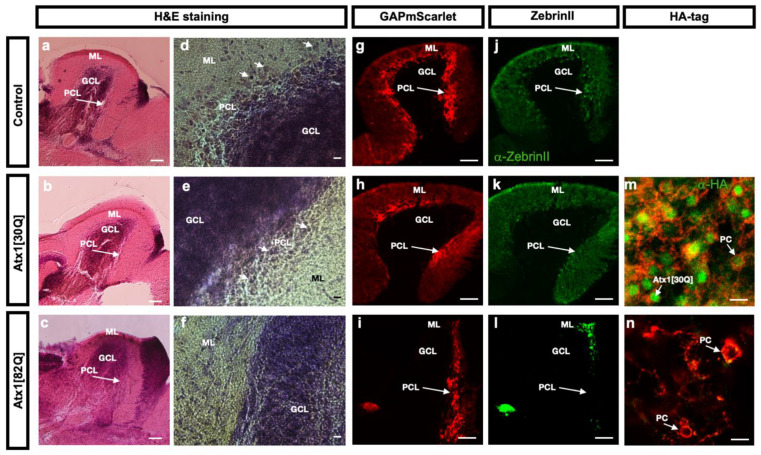
Histological comparison of adult corpus cerebelli of control and *hAtx1* expressing zebrafish. Sagittal cryogenic sections through the corpus cerebelli of three months old adult zebrafish from transgenic GAPmScarlet controls (upper row), Atx1[30Q] (middle row) and Atx1[82Q] (lower row) heterozygous carriers, anterior is to the left. (**a**–**c**) Hematoxilin and Eosin staining reveals no changes in gross morphology of the corpus cerebelli in overview images, yet at higher magnifications (**d**–**f**) large somata (white arrows) characteristic for PCs juxtaposed to the GCL are only visible in control and Atx1[30Q] cerebelli but are hardly found in Atx1[82Q] cerebelli. This absence of PCs is further supported by different patterns of red GAPmScarlet fluorescence, which in controls (**g**) and Atx1[30Q] (**h**) samples appears as a continuous layer (white arrow), but can be found in cerebellar tissue of Atx1[82Q] carriers (**i**) only in posterior regions which is further confirmed by green fluorescent immunohistochemistry using the PC-specific ZebrinII-antibody (**j**–**l**). Green fluorescent anti-HA-tag immunohistochemistry confirms the maintenance of human Atx1-protein expression in PCs, yet compared to Atx1[30Q] cerebelli (**m**) only with sparse labelling in PCs (white arrows) of Atx1[82Q] carriers (**n**) suggesting a progressive state of degeneration. Scale bars: 100 µm (**a**–**c**,**g**–**l**), 10 µm (**d**–**f**,**m**,**n**). Abbr.: GCL: granule cell layer, ML: molecular layer, PCL: Purkinje cell layer.

**Figure 5 ijms-22-07351-f005:**
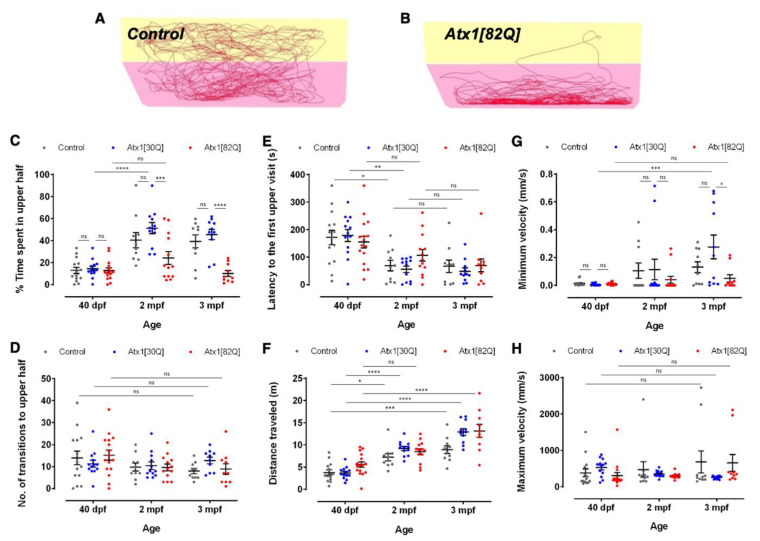
Age-dependent exploratory behavioural comparison of control and *hAtx1* expressing zebrafish. Novel tank diving tests were performed with GAPmScarlet, Atx1[30Q], and Atx1[82Q] larvae at 40 dpf, juveniles at two months and young adults at three months of age, respectively. Examples of 6-min traces for (**A**) a GAPmScarlet control fish and (**B**) an Atx1[82Q] carrier are shown, the bottom half of the tank is marked in pink, the upper half in yellow. (**C**) Percentage of time spent in the upper half of the novel tank during a 6-min observation period. (**D**) Number of entries into upper tank half during the entire observation period. (**E**) Latency until upper tank half is entered for the first time. (**F**) Total distance travelled during the entire observation period. (**G**) Minimal and (**H**) maximal velocity of swim movements measured during the observation period. Data are presented as mean ± SEM, * *p* < 0.05, ** *p* < 0.01, *** *p* < 0.001 and **** *p* < 0.0001 (according to two-way ANOVA with post hoc Tukey’s test). *n* = 14,10,10 in 40 dpf, 2 mpf, and 3 mpf control groups; *n* = 13, 12, 11 in 40 dpf, 2 mpf and 3 mpf Atx1[30Q] groups; and *n* = 15, 13, 10 in 40 dpf, 2 mpf, and 3 mpf Atx1[82Q] groups, respectively.

## Data Availability

Data and tools described in this manuscript are available upon request.

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
