# Peer review of "Genetic Modeling of the Neurodegenerative Disease Spinocerebellar Ataxia Type 1 in Zebrafish"

_ijms, 2021, doi:10.3390/ijms22147351_

Round 1
Reviewer 1 Report
In the manuscript, the authors made a genetic modeling system of Spinocerebellar Ataxia Type 1 (SCA1) by cell type-specific overexpression in zebrafish. They characterized the model system by analyzing age-dependent PC degeneration and behavioural performance of disease model. Although the manuscripts is generally well-written and experiments are well designed. I have few considerations, the authors should adress.
In the result part (2.1 Genetic modeling of SCA1 in zebrafish), the authors summarized the establishment of the disease model. However, they should explain in more detail for the reader. For example, they mentioned as a control, they used cDNA of human ataxin1 containing a non-pathogenic polyglutamine stretch of 30 glutamine residues Atx 1(30Q) interrupted by two Histidine residues. The authors should explain further the reason for selection of this interruption by two histidine residues in non-pathogenic polyglutamine stretch of 30 glutamine residues.
Moreover, the authors should discuss the advantage of this model against previously established zebrafish models of ataxia 1(PMID: 33671313 ).
Reviewer 2 Report
My comments:
- Is there any difference between zebrafish SCA1 model, compared to mouse models? What are the benefits and disadvantages of zebrafish models in SCA1, compared to mouse/fruit fly models?
- Were there any other ataxias (neurodegenerative diseases) modeled in zebrafish?
- I would add a figure on pathway, how SCA1 zebrafish model was made
- I would also add a pathway in which ATXN1 polyQ expansion caused in zebrafish models. Is it similar pathway. which was seen in mouse models?
Round 2
Reviewer 2 Report
Authors fulfilled my suggestions